# Evaluation of Three Different Approaches for Automated Time Delay Estimation for Distributed Sensor Systems of Electric Vehicles

**DOI:** 10.3390/s20020351

**Published:** 2020-01-08

**Authors:** Jakob Pfeiffer, Xuyi Wu, Ahmed Ayadi

**Affiliations:** 1BMW Group, Petuelring 130, 80788 Munich, Germany; 2Department of Electrical and Computer Engineering, Technical University of Munich, Arcisstr. 21, 80333 Munich, Germany

**Keywords:** automotive, current, electric power train, electric vehicle, embedded systems, delay, detection, distributed systems, measurements, power train, sensor, signals, time delay estimation

## Abstract

Deviations between High Voltage (HV) current measurements and the corresponding real values provoke serious problems in the power trains of Electric Vehicle (EVs). Examples for these problems have inaccurate performance coordinations and unnecessary power limitations during driving or charging. The main reason for the deviations are time delays. By correcting these delays with accurate Time Delay Estimation (TDE), our data shows that we can reduce the measurement deviations from 25% of the maximum current to below 5%. In this paper, we present three different approaches for TDE. We evaluate all approaches with real data from power trains of EVs. To enable an execution on automotive Electronic Control Unit (ECUs), the focus of our evaluation lies not only on the accuracy of the TDE, but also on the computational efficiency. The proposed Linear Regression (LR) approach suffers even from small noise and offsets in the measurement data and is unsuited for our purpose. A better alternative is the Variance Minimization (VM) approach. It is not only more noise-resistant but also very efficient after the first execution. Another interesting approach are Adaptive Filter (AFs), introduced by Emadzadeh et al. Unfortunately, AFs do not reach the accuracy and efficiency of VM in our experiments. Thus, we recommend VM for TDE of HV current signals in the power train of EVs and present an additional optimization to enable its execution on ECUs.

## 1. Introduction

Political guidelines in various countries to decarbonize individual mobility led to an exponential growth of EVs in offers and sales. However, one obstacle for the success of EVs is the so-called range anxiety [1]. Customers are afraid that an EV is not able to provide the range they need for all of their journeys. To combat range anxiety and increase the range of EVs, there are two different ways. The first one is to simply increase the size of the High Voltage Battery (HVB). Unfortunately, this means to increase the size of the most expensive component of an EV, and after all, it is not a very sustainable way. The second way, which is our solution of choice, is to make EVs more efficient.

*Kirchhoff’s current law* states that the sum of all currents at a node of an electric system is equal to 0 A. However, considering measurement signals of nodes in the power trains of EVs with distributed sensor systems, the sum of all currents can differ up to 20% of the maximum current (see Figure 1). If we look closer at the Root Mean Square Error (RMSE) of the sum of currents $RMSE(i_{\mathrm{sum}})=0.67\%$, we realize that it has the same value as the mean current of the DCDC converter $\mu_{i_{\mathrm{DCDC}}}=0.67\%$.

A different value than 0 A for the sum of all currents indicates that there is a divergence between measurements and real values. The divergence becomes problematic when the power train is operating close to the system boundaries. For example, there are boundaries for the protection of the HVB. The HVB is only capable of discharging or charging a restricted amount of power. Higher amounts would threaten the HVB’s lifetime and safety [2]. To ensure a safe operation mode even for high divergences between measurements and real values, additional protection offsets (see Figure 2) might be added to the boundaries, although they have some drawbacks.

For example, in the charging case, most notably during recuperation, the HVB might not allow the full power level, even though it would be capable of handling it. Thus, the amount of power charged to the battery is restricted and the EV loses cruising range while its power consumption increases. In the opposite case, the system might not release requested power, although the HVB could provide it in reality. This additional restriction of power decreases the EV’s performance. As can be seen from the two examples above, minimizing the magnitude of the protection offsets also allows increasing the performance as the efficiency and the cruising range of EVs.

Besides measurement faults and sensor uncertainties [3], the divergence between measurements and real values is caused by time delays. Figure 3 shows an example of the sum of all currents $i_{\mathrm{sum}}$ being reduced by shifting a signal by 6 time steps. The delays result from distributed sensor systems in the power train as plotted in Figure 4. The High Voltage (HV) components have their own ECU which is connected with the current sensors and processes the sensor information. The ECUs exchange this information via bus systems. The buses require individual amounts of time to send the measurement signals. Thus, from an ECU’s point of view, the sensor information from other ECUs arrives with individual delays (see the Ego ECU in Figure 4). These individual delays could be compensated easily with a synchronized clock and time stamps as part of each bus message. However, this solution would have two drawbacks. First, it would increase the bus traffic as not only the measurement information must be carried by the messages but also the time stamp. As a result, the EV would either require a faster bus which is able to transport more information, or it would have to reduce the information exchanged between the ECUs. Second, there exists no clock in the power trains of modern series EVs which is synchronized with all ECUs at the same frequency as the message exchange. Usually, the ECUs are synchronized in a longer time frame than they communicate. Thus, the time stamp solution would require additional or higher performing hardware and increase the costs for the production of the EV.

The aim of this work is to automatically detect the time delay between measurement signals from different sensors without additional hardware. For this purpose, we develop two different approaches. One of them is based on LR, whereas the other one optimizes the estimated variance of the difference between several signals. We compare our approaches to other state-of-the-art TDE algorithms and evaluate them with a focus on precision and run-time efficiency. Apart from allowing a more accurate power distribution, the automated TDE helps to reduce the battery protection offset and thus to increase the performance, efficiency and cruising range of EVs.

The rest of this paper is structured as follows. Section 2 states related work and the similarities and differences to our work. Furthermore, Section 2 highlights the contributions of our work to the state of the art. In Section 3, we explain the theory behind our work before we describe the practical experiments in Section 4. The experiments’ results, stated in Section 5, show us the performance of the algorithms for our use case. Based on this evaluation, we take the best performing algorithm and optimize it further. The optimization steps can be taken from Section 3.4 and their impacts to the results from Section 5.4. In Section 6, we discuss the advantages and drawbacks of all proposed concepts. Finally, we draw our conclusions and give a short outlook in Section 7.

## 2. State of the Art

There exists plenty of literature about TDE, although—to the best of our knowledge—none of them is tailored to the specific problem of TDE of current signals in EVs. In the following, we present several publications about TDE from different fields of application, such as embedded systems, acoustics, medicine, positioning, aeronautics, process technology, and robotics.

An approach which also deals with EVs and time delays is the one by Guo et al. [4]. However, their approach is similar to ours only at the first look. Their goal is to stabilize a grid of electric sources and sinks with EVs. For the stabilization of the grid, they propose time delay resistent control strategies of smart grids with EVs. The EVs are able to charge bidirectionally. The bidirectional charging is used to smooth disturbances and respond rapidly to fast occurring changes in the power distribution of the grid. An example for such a rapidly occurring change in the times of renewable energies is the power output of wind turbines when a strong wind occurs. Compared to our approach, Guo’s focus is rather on the control strategy than on the TDE. Another difference with our work is that Guo’s system is rather macroscopic with lots of different elements and many EVs in the grid. Our system is instead quite microscopic. We consider a single EV with a power train of around five sources and sinks. Our communication network might be smaller than the number of HV components as some consumers might share the same ECU. For example, the heating and the cooling component of an EV use both the climate control ECU for bus communication.

Kali et al. [5] propose a controller with TDE for Electric Machine (EMs). The TDE is executed state-based with the help of a model of the EM. The model design demands expert knowledge about the physical principles of an EM. This is justified for Kali et al. as they require the same knowledge for their controller. However, for our case, we want to be able to estimate the time delays without further knowledge about the HV components. Our TDE shall be executable with nothing else than the available measurement data.

Zeng et al. [6] introduce a statistical approach to predict the delay of a bus message. The content of the messages does not need to be known to achieve high accuracy. This is different from our scenario where we want to make use of the information carried by the message. In contrast to Zeng et al., we do not require predicting the time delay accurately to milliseconds. For our purposes, an estimation of the number of delayed discrete time steps is sufficient.

Not from the field of electric power trains or bus communication, but from acoustics is the approach shown by Lourtie and Moura [7]. They use a stochastic approach to model time delays in an acoustic path environment. Like ours, their environment consists of several sources. However, in contrast to our scenario, the delay they want to estimate varies with time. In our case, we assume the time delay to be constant in a short time frame. For longer periods, it might change slowly. The reason for the slowly changing time delay is that it is caused during the wake up procedure of the EV. The ECUs wake up in an unsynchronized way. Afterwards, the ECUs are synchronized on a relatively large time frame (e.g., 1 s), but work based on short time steps (e.g., 10 ms).

Another acoustics application for TDE is shown by He et alii [8]. They use the so-called Multichannel Cross-Correlation Coefficient algorithm to estimate time delays of speech sources in noisy and reverberant environments.

Svilainis et al. [9] present another interesting approach. Their goal is to estimate the time passed between emitting an ultrasonic signal and absorbing its reflection. Like Zeng et al., they require high precision. Another difference to our approach is that their algorithms make use of the pulse form of ultrasonic signals. Our signal as plotted in Figure 1 can vary in a large range and does not necessarily contain pulses (e.g., after time step 5,000).

Mirzaei et al. expand TDE for ultrasonic signals to the field of medicine [10]. The authors introduce a window-based TDE approach to estimate the time passed between two frames of radio-frequency data. They compare the results of the new window-based approach to their previously developed, optimization-based method [11] and to Normalized Cross-Correlation.

Recently, Garcez et al. published their work on a similar problem to ours, but in a completely different field of application [12]. Like bus systems of EVs, Global Navigation Satellite Systems (GNSSs) systems have real-time requirements. Their goal is to minimize deviations between measurements and real position data. The time delays are caused during the transmission of GNSS messages, when the signals do not take straight lines of sight, but are reflected on their way or suffer from noise. The authors propose a tensor-based subspace tracking algorithm to efficiently estimate time delays of received GNSS signals.

A similar approach is presented by Xie et al. for an indoor positioning sensing system [13]. They sense positions of mobile devices based on the signal strength and the signal’s time delay since its transmission from a base station. For the TDE, Xie et al. combine Cross-Correlation with Quadratic Fitting. This is similar to our LR approach (see Section 3.2), where we try to fit the signals with quadratic functions to retrieve the delay between them. Like Garcez et al., they have to deal with the problem that the signals are often reflected and do not take direct lines of sight. Different to Garcez et al., Xie’s approach takes the strength of the signal into account for retrieving a more exact position estimation. For our work, we cannot take advantage of this information, because in wire-based bus systems all signals are equally strong.

Schmidhammer et al. estimate positions of moving, non-cooperative objects in vehicular environments [14]. Their idea is to estimate the position of an object based on time delays in a network of distributed receiving and transmitting nodes. In contrast to our work, the networking nodes of Schmidhammer et al. are not necessarily on-board the vehicle, but can also be mounted on the road infrastructure.

Emadzadeh et al. [15] show an inspiring approach for detecting the relative position of spacecrafts. For retrieving the position, they examine an X-ray signal received by two spacecrafts and determine the time delay between them. For the TDE, they use AFs. This approach seems very promising to us. We implement the algorithms of Emadzadeh et al. and compare them to ours in order to find out if their approach can be transferred from X-ray signals to current measurements in the power train of EVs.

Like Emadzadeh et al., Liu et al. focus on AFs [16]. Compared to our problem of fixed or only slowly changing time delays, the difference in Liu et al. is that they deal with time-varying time delays. That makes further processing steps necessary. For example, they require a transition probability matrix and an initial probability distribution vector to model the time delay changes with a Markov chain.

Park et al. analyze time series data with Autoencoders and Long Short-Term Memory Neural Network (LSTMs) to detect faults in industrial processes [17]. The authors emphasize the importance of TDE for correct fault detection. However, they focus only on time delays caused by their own fault detection system. Our focus lies on earlier steps in the processing chain. We want to detect time delays between the input signals before they are passed to other computation processes. Furthermore, we want to implement algorithms which are able to learn on-board the automotive ECUs and adapt themselves to new data. As the training of Neural Networks is quite memory intensive and demands high computational power, they do not belong to our methods of choice.

Close to the application field of industrial processes is the approach of Srinivasa Rao et al. [18]. In their recent article, the authors propose fuzzy parametric uncertainty to mathematically model systems with time delays. Their goal is to enable a robust controller design. For this purpose, they first approximate the time delay system as an interval system. After retrieving the intervals, they design an optimal controller for these. Like Guo et al., Srinivasa Rao et al. focus on how to retrieve an optimal controller, which is not part of our work. Although they focus on the control of industrial processes, their article is very general. Besides industrial plants, they also mention potential fields of application, such as EMs or robot manipulators.

Time delay compensation for robots is the focus of Shen et al [19]. Their focus is on teleoperating robots which require knowledge about the time delay between the master and the slave robot for stable operation. The robots and their communication channels are modeled as extended dynamical system. For this system, Shen et al. develop a cascade observer which is able to control it in a stable way. The authors assume that a sufficiently accurate value for the TDE is given and concentrate on its compensation. This is different to our work here. We explicitly want to estimate the time delay.

You et al. develop a proportional multiple integral observer for fuzzy systems [20]. The goal of their work is the same as ours. They want to minimize deviations between measurements and real values caused by time delays and measurement inaccuracies. Their time delays are also varying. Unlike the varying time delays presented before, the ones of You et al. do not vary with time but rather with states. Their focus is also on industrial processes and not on electric power trains. However, the main difference between our works is that You et al. want to minimize time delays and measurement inaccuracies with the same system.

Our approach follows the *divide and conquer* strategy and faces the two problems separately. We focus on the problem of measurement deviations caused by measurement inaccuracies in our previous work [3]. However, measurement inaccuracies are not part of this work. Here, we assume that the measurements are appropriately accurate and that the main deviations are caused by time delays as shown in Figure 1 and Figure 3. Thus, TDE is our solution of choice to minimize the deviations.

Our contribution in this article is the development of a regression-based approach and an algorithm based on VM for TDE as first presented in [21]. We transfer the ideas introduced by Emadzadeh et al. to the domain of currents in the HV system of EVs and compare the results to our approaches in matters of accuracy and computational performance. Our TDE works only with the data available in modern series EVs and does not require an additional clock. In addition to [21], we introduce an optimization of the most accurate and efficient of our evaluated approaches. We further evaluate the optimization both on artificially created data with known ground truth as well as real drive data with unknown ground truth.

## 3. Concepts

In this section, we introduce the algorithms and shortly explain the concepts from other authors which we implement and compare for TDE. From now on, for the sake of easier understanding, we focus on the current of the EM $i_{\mathrm{EM}}$ and the HVB $i_{\mathrm{HVB}}$ (without other consumers than the EM) as examples. Nevertheless, the proposed methods can be extended to every current signal in the HV system of an EV. Furthermore, we inverse the sign of $i_{\mathrm{HVB}}$ from now on to make its shape similar to the one of the EM. Thus, we can treat the HVB current signal as a delayed or preceded version of the EM, respectively.

Our goal is to find the time delay $t_{d}$ in a bus system which can be described as
(1)$$\begin{array}{cc} x_{1}\left(t\right) & =i_{1}\left(t\right)+n_{1}\left(t\right)\\ x_{2}\left(t\right) & =i_{2}\left(t-t_{d}\right)+n_{2}\left(t-t_{d}\right), \end{array}$$
where *t* stands for the time step, $x_{1}\left(t\right)$ is the measurement signal of the faster component, $x_{2}\left(t\right)$ describes the slower component’s signal, $i_{1}\left(t\right)$ and $i_{2}\left(t\right)$ describe the corresponding currents and $n_{1}\left(t\right)$ and $n_{2}\left(t\right)$ are noise terms [15]. As we cannot retrieve the currents $i_{1}\left(t\right)$ and $i_{2}\left(t\right)$ directly, we cannot minimize the difference between $i_{1}\left(t\right)$ and $i_{2}\left(t\right)$. Instead, we directly minimize the difference between the two measurement signals $x_{1}\left(t\right)$ and $x_{2}\left(t\right)$.

### 3.1. Adaptive Filter

The idea of Emadzadeh et al. is to model the time delay as Finite Impulse Response (FIR) filter. They define $x_{1}\left(t\right)$ to be the faster signal. For each measurement $x_{2}\left(t_{i}\right)$ at time step $t_{i}$, they collect a row of the last *M* measurements of the other signal
(2)$$x_{1}\left(t_{i}-M+1:t_{i}\right)=\left[x_{1}\left(t_{i}-M+1\right),x_{1}\left(t_{i}-M+2\right),\cdots x_{1}\left(t_{i}-1\right),x_{1}\left(t_{i}\right)\right].$$
Then, the authors search for an optimal channel impulse response vector $\omega^{*}$ such that the deviation between $x_{2}\left(t_{i}\right)$ and $x_{1}\left(t_{i}-M+1:t_{i}\right)\omega^{*}$ becomes minimal. Mathematically, this can be expressed by the minimization of the expectation value of the Mean Squared Error (MSE) between the measurement value of the slower signal and the filtered measurement row of the faster signal. It results in the formula
(3)$$\omega^{*}=\underset{\omega }{\mathrm{argmin}\;}E\left[{\left(x_{2}\left(t_{i}\right)-x_{1}\left(t_{i}-M+1:t_{i}\right)\omega \right)}^{2}\right].$$
This is similar to our VM approach (see Section 3.3) with the difference that we minimize the variance instead of the MSE. In Emadzadeh’s work, the optimal factor $\omega^{*}$ is estimated recursively. For the recursion, the authors implement and compare the four algorithms Least Mean-Squares (LMS), Normalized Least Mean-Squares (NLMS), Least Mean-Fourth (LMF) and Recursive Least-Squares (RLS). The optimal time delay estimate $t_{d}^{*}$ is then the one where the impulse response $\omega^{*}$ reaches its maximum, or mathematically
(4)$$t_{d}^{*}=\underset{i\in \left[1,M\right]}{\mathrm{argmax}\;}\omega^{*}\left(i\right)-1.$$

For further details on this approach, we kindly refer the interested reader to [15].

### 3.2. Linear Regression

Our first approach is to use LR to identify the *basis functions* of two received signals and compare the horizontal offset between these functions. As degree of the basis function, we choose a parabola for two reasons. First, the sampling frequency of our measurements is high enough to fit the signals with a parabola for a short time duration. Second, the comparison of the horizontal offset is easiest with a parabola because it only has one extremum.

We collect the last *M* measurements $x_{k}$ of the HV components k∈{EM,HVB} in a measurement vector $y_{k}\;=\;\left(\begin{array}{cccc} x_{k}\left(t\right) & x_{k}\left(t-1\right) & \cdots & x_{k}\left(t-M+1\right) \end{array}\right)$. Then, we retrieve the weight vector $w_{k}\;=\;{\left(\begin{array}{ccc} w_{k,0} & w_{k,1} & w_{k,2} \end{array}\right)}^{T}$ with LR [22] according to
(5)$$w_{k}={\left(\sum_{n=1}^{N}\phi^{n}{\left(\phi^{n}\right)}^{T}\right)}^{-1}\sum_{n=1}^{N}y_{k}^{n}\phi^{n}.$$
Here, the notation $y_{k}^{n}$ and $\phi^{n}$ represents the *n*-th column of $y_{k}$ and ϕ, respectively. The so-called *design matrix*
$$\phi ={\left(\begin{array}{ccc} 1 & t & t^{2}\\ 1 & t-1 & {\left(t-1\right)}^{2}\\ \vdots & \vdots & \vdots \\ 1 & t-M+1 & {\left(t-M+1\right)}^{2} \end{array}\right)}^{T}$$
consists of N=3 columns in our case.

With the weight vector from (Equation 5), we are able to fit a parabola
(6)$$f_{k}\left(t\right)=w_{k,0}+w_{k,1}t+w_{k,2}t^{2}$$
as basis function to the measurement vector $y_{k}$.

After retrieving the basis functions in (Equation 6), we transfer them into vertex form
(7)$$f_{k}\left(t\right)=w_{k,2}{\left(t-x_{k,\mathrm{vertex}}\right)}^{2}+y_{k,\mathrm{vertex}}$$
to identify the coordinates $\left(x_{k,\mathrm{vertex}},y_{k,\mathrm{vertex}}\right)$ of each basis function’s vertex. The estimated time delay between the EM and the HVB current signals is then given by the difference on the time axis between their vertices according to
(8)$$t_{d}^{*}=x_{\mathrm{EM},\;\mathrm{vertex}}-x_{\mathrm{HVB},\mathrm{vertex}}.$$

### 3.3. Variance Minimization

Our second approach is to minimize the variance of the difference between two signals $x_{1}\left(t\right)$ and $x_{2}\left(t\right)$ by shifting the signal $x_{2}\left(t\right)$ forward.

Like in Section 3.2, we collect the two signals $x_{1}\left(t\right)$ and $x_{2}\left(t\right)$ for *M* time steps. A straightforward idea for the minimization of the difference between $x_{1}\left(t\right)$ and $x_{2}\left(t\right)$ is to minimize their estimated MSE
(9)$$\mathrm{MSE}\left(x_{1}\left(t\right),x_{2}\left(t\right)\right)=\frac{1}{M-T_{\max}}\sum_{t=1}^{M-T_{\max}}{\left(x_{1}\left(t\right)-x_{2}\left(t+t_{d,i}\right)\right)}^{2}$$
with different time shifts $t_{d,i}$ in a pre-defined range $t_{d,i}\;\in \;\left[T_{\min},T_{\max}\right]$ with $T_{\max}<M$. However, our experiments show that we need a relatively high *M* to achieve stable results. We can significantly minimize *M*, if we take the estimated expected value
(10)$$E=\frac{1}{M-T_{\max}}\sum_{t=1}^{M-T_{\max}}\left(x_{1}\left(t\right)-x_{2}\left(t+t_{d,i}\right)\right).$$
into account. Thus, instead of minimizing the MSE from (Equation 9), we minimize the estimated variance of the difference between the two signals
(11)$$\sigma^{2}=\frac{1}{M-T_{\max}}\sum_{t=1}^{M-T_{\max}}{\left(\left(x_{1}\left(t\right)-x_{2}\left(t+t_{d,i}\right)\right)-E\right)}^{2}.$$

The time delay between the EM and the HVB current signals is the $t_{d,i}$ that minimizes the variance
(12)$$t_{d}^{*}=\underset{t_{d,i}}{\mathrm{argmin}\;}\sigma^{2}.$$

As we do not know in the beginning whether $x_{\mathrm{EM}}\left(t\right)$ or $x_{\mathrm{HVB}}\left(t\right)$ is the faster signal, we have to choose one of them as $x_{1}\left(t\right)$ and the other one as $x_{2}\left(t\right)$ for the first execution and try $T_{\min}=-T_{\max}$. From the second execution on, the value of $T_{\min}$ and $T_{\max}$ can be reduced and chosen recursively, because the EV’s bus system usually changes its time delay only once in the beginning, but not during advanced execution. Therefore, we choose $T_{\min}\left(t\right)=t_{d}\left(t-1\right)-1$ and $T_{\max}\left(t\right)=t_{d}\left(t-1\right)+1$ from the algorithm’s second execution on.

### 3.4. Optimized Variance Minimization

As the results of our experiments (see Section 5) show, the VM concept provides the best results in terms of RMSE, run-time and required frame size. However, when running this concept in real time (both on simulated and real data, see Section 5.4), we find that the TDE is unstable and that the estimated time delay frequently alternates between different values. These many changes of the estimated time delay contradict the fact that the time delay is rather stable in reality, and that, if any, changes occur after relatively long periods. Thus, in order to stabilize the TDE, we suggest the following improvement of the *plain* VM approach from Section 3.3.

The main idea of the stabilization is to use a statistical test. The test’s purpose is to quantify the *reliability* of the input data segment on which the TDE is performed. In fact, we know from Section 3.3 that the estimated time delay $t_{d}$ at time step *t* minimizes the variance given by Equation (Equation 11). This equation in turn is based on the *M* last values of both signals. Due to the noise in the data, the estimation for the next step can jump to a different value, even if the vast majority of data points (M−1) are shared between the two steps. The idea is thus to compare at each step the minimal variance with the second smallest one. If the difference between both in relative terms is not sufficiently large, we presume that the TDE is not reliable, and consequentially do not estimate a time delay. In this case, we simply keep the prediction from the last step. Otherwise, we update the estimation to the newly calculated $t_{d}$.

Formally, at each time step, we calculate the variance criterion from Equation (Equation 11) for each potential time delay $t_{d,i}$. Let us denote this by $\sigma^{2}\left(t_{d,i}\right)$. Then, we know that the least achievable variance is given by
(13)$$\sigma^{2}\left(t_{d}^{*}\right)=\underset{t_{d,i}}{\min\;}\sigma^{2}\left(t_{d,i}\right).$$

The second smallest achievable variance in turn is given by
(14)$$\sigma^{2}\left(t_{d}^{**}\right)=\underset{t_{d,i}\neq t_{d}^{*}}{\min\;}\sigma^{2}\left(t_{d,i}\right).$$

In other words, we minimize the variance over all potential time delays except that which minimizes it ($t_{d}^{*}$). By definition, we have $\sigma^{2}\left(t_{d}^{**}\right)\geq \sigma^{2}\left(t_{d}^{*}\right)$. The intuition is that if the difference between those two values is not large enough, the noise makes it impossible to tell with high confidence which one is the real minimizer. The minimum in Equation (Equation 13) might result in $t_{d}^{*}$ by random noise instead of being the true minimum. Thus, we suggest deciding whether to perform an update based on the criterion
(15)$$\frac{\sigma^{2}\left(t_{d}^{**}\right)-\sigma^{2}\left(t_{d}^{*}\right)}{\sigma^{2}\left(t_{d}^{*}\right)}>K,$$
where *K* is a hyper-parameter defining the minimal percentage error required to perform an update. Clearly, the larger *K*, the more severe is the criterion, and the fewer updates are done. Therefore, we choose *K* to strike a balance between reliability on the one hand, and being up-to-date on the other hand. In fact, if we choose *K* too high, updates are performed only rarely, so that we can miss changes in the underlying real time delay. If *K* is chosen too small, then the predictions are more unstable. We empirically found K=0.2 to strike a balance between both criteria for our power train data.

## 4. Experimental Setup

In this section, we explain the data and the setup for the experiments to evaluate the performance of the three concepts for TDE and the optimization presented above.

### 4.1. Data

For the evaluation of the three concepts and the optimization shown in Section 3, we use 74 data sets. The data sets contain all currents of the HV system and are recorded during representative drives on public roads with close to production EVs. We use bus loggers to record the data. The loggers store the received measurement signals from all ECUs and write them to a log file during each time step. After driving, we use the log files to execute our experiments and evaluate our approaches. Thus, the algorithms get at each time step the same input data which they would receive during execution on an ECU in the real EV. The recordings correspond to 10 h 33 min of driving. For the experiments, the 74 data sets are divided into 409 sub-data sets with a maximum length of 10,000 time steps. The minimum length among the 409 sub-data sets is 1,807 time steps.

For the Optimized VM approach, we create an additional data set artificially. The artificial data set bases upon the real data sets described above. However, instead of computing the time delay between two real signals, we introduce an artificial signal. This artificial signal is a real signal shifted by some time steps. We can then compute the time delay between the original signal and its artificially delayed correspondence. This has the advantage that we exactly know the time delay and thus know the ground truth.

### 4.2. Experiments

According to *Kirchhoff’s current law*, we assume that the sum of the measurements of *i*_HVB_, *i*_EM_, *i*_heat_, *i*_cool_ and *i*_DCDC_ is zero. Thus, we estimate the *ground truth* of the time delay for each real data set by minimizing the MSE of the complete data set (see Equation (Equation 9)). In this case, *M* is the length of the data set, $T_{\max}$ is 10 time steps and $T_{\min}$ is −10 time steps, since the time delay in the EV is normally smaller than ten time steps.

In the experiment of the AF concept, we choose the frame size of $2^{10}$ as proposed by [15]. Additionally, we execute our experiments with a more efficient frame size of $2^{8}$. We assume the length of the AF to be 10. All the other parameters for the used algorithms are chosen equally to [15]. Furthermore, we evaluate five different learning rates for LMS.

For the evaluation of the VM concept we test different frame sizes M∈{30,50,100,200,300}. In the first calculation, we also choose $T_{\max}=10$ and $T_{\min}=-10$. From the second execution on we select $T_{\min}\left(t\right)=t_{d}\left(t-1\right)-1$ and $T_{\max}\left(t\right)=t_{d}\left(t-1\right)+1$.

For the Optimized VM, we choose a fixed frame size of 50 time steps. This frame size proved to be the best compromise between run-time and accuracy in previous experiments as described in Section 5.3.

For each of the three concepts, we calculate the time delay every 20th time step. In total, this results in around 90,000 time delay estimations for each concept.

### 4.3. Environment

All concepts are implemented in Matlab R2015b with Microsoft Windows 10 on an HP^®^ EliteBook™840 G3 with an Intel^®^ Core™i5-6300U 2.40GHz CPU and 8 GB RAM.

## 5. Results

In this section, we present the results of our experiments and evaluate the performance of the three concepts and the optimization individually. The results of all three algorithms compared next to each other can be found in the next section.

### 5.1. Adaptive Filter

Based on the learning rate and parameters in [15], the RLS algorithm performs better than the LMS, NLMS and LMF algorithms (see Table 1).

Furthermore, we analyze the learning rate for the LMS algorithm. As mentioned in [15], the learning rate μ is typically chosen in the range $0<\mu <2/(M\sigma_{u}^{2})$, where $\sigma_{u}^{2}$ is the input signal variance and *M* is the length of the filter. Thus, we compare the performance of LMS with different $\mu =a/(M\sigma_{u}^{2})$ and a∈{0.01,0.05,0.1,0.5,1}. In Table 2, the RMSE has the minimal value of 2.1479 with a=0.1. It is much smaller than the RMSE of 2.8972 with the fixed learning rate in [15]. For a too large or a too small *a*, the performance of the LMS decreases significantly. This result is expected, since a too small learning rate leads to slow convergence while a large one most often misses the optimum.

In addition, we evaluate the LMS algorithm with a more efficient frame size of $2^{8}$. As printed in Table 3, the smaller frame size improves the run-time. Although some non-optimal learning rates improve their estimation accuracy, which we explain with the drop of local minima due to the shortened frame, the two best learning rates in the experiment with the frame size of $2^{10}$ increase their estimation errors with the smaller frame size.

### 5.2. Linear Regression

Our first approach LR is, to a large extent, affected by noise and the offset between the two signals caused by measurement inaccuracies. Especially this offset leads to an imprecise estimation of the vertices and thus a wrong estimated time delay. Figure 5 shows an example for such a wrong estimation. In this data set, the time delay between *i*_HVB_ and *i*_EM_ is equal to 6 time steps. We train both curves on 200 measurement samples of their corresponding signal. However, due to noise and some vertical offset between the signals the vertex of the slower signal is not only shifted to the right but also to the top. The shift in vertical direction also affects the horizontal position of the vertex and results in a wrong TDE of 43 time steps.

Table 4 shows the results and the average run-time of this approach with three different frame sizes. The run-time grows with increasing frame sizes, whereas the RMSE becomes smaller. Nevertheless, the RMSEs are in general very high even for large frames.

### 5.3. Variance Minimization

Table 5 shows the RMSE between the ground truth of the time delay and the calculated time delay. Furthermore, the table shows the average of the run-time for each time delay calculation, corresponding to different frame sizes *M* (in Equation (Equation 11)). We see that the concept requires a relatively short run-time as it benefits from the recursive calculation only in the area $t_{d,i}\;\in \;\left[T_{\min},T_{\max}\right]$ with $T_{\min}\left(t\right)=t_{d}\left(t-1\right)-1$ and $T_{\max}\left(t\right)=t_{d}\left(t-1\right)+1$. In addition, the RMSE decreases while the size of the frame increases. The accuracy has a large improvement when the frame is enlarged from 30 time steps to 50 time steps.

### 5.4. Optimized Variance Minimization

We evaluate the proposed stabilization approach twofold. First, we evaluate it based on simulated data with known ground truth time delay. Second, we evaluate the approach on real signals. While the first experiment shows the accuracy of the proposed approach, the second one shows its effectiveness in providing more stability.

#### 5.4.1. Evaluation with Simulated Signals

In this experiment, we first take a current signal $x_{1}\left(t\right)$ from a real-world data set recorded on-board of an EV. Based on $x_{1}$, we then create a second signal $x_{2}\left(t\right)=x_{1}\left(t-t_{d}\left(t\right)\right)+n_{2}\left(t\right)$. Therefore, the ground-truth time delay $t_{d}\left(t\right)\geq 0$ is a realization of a random jump process that in known in advance. Furthermore, $n_{2}\left(t\right)$ is a white noise process whose variance is chosen such that the resulting Signal-to-Noise Ratio (SNR) is equal to -10. We then run our VM algorithm with and without stabilization to detect the delay $t_{d}\left(t\right)$. Figure 6 shows the results of this experiment.

#### 5.4.2. Evaluation with Real Signals

In this experiment, we take both signals $x_{1}\left(t\right)$ and $x_{2}\left(t\right)$ from a real-world data set. We then run the VM approach with and without stabilization, and plot the results in Figure 7.

## 6. Discussion

We discuss the advantages and drawbacks of the previously described and evaluated concepts in this section.

Although it is not as efficient and accurate as VM, the AF approach still retrieves better results than LR. The best results for AFs, in our case, are reached with the LMS algorithm and a learning rate of $\mu =0.1/(M\sigma_{u}^{2})$ (see Table 2). The learning rate, which must be chosen manually, is one drawback of this algorithm. It can lead to sub-optimal learning if the user chooses a wrong value. In contrast with LMS, the RLS algorithm does not require a learning rate. However, we see that the RLS algorithm has lower accuracy, requires longer run-time and more memory for a larger frame than the VM concept (see Table 6).

LR has the advantage that it can directly find out the faster component. Thus, one single execution during the same time step for the same signal is sufficient even in the beginning, which makes it interesting, if a computational effective approach is needed. However, its efficiency suffers from the matrix inversion in Equation (Equation 5). Even worse, it is the least accurate of the three proposed concepts due to noise and vertical offsets between the signals. The high estimation errors make this approach unfeasible for our purpose. Another drawback is that a parabola is not always the optimal basis function for the regression of measurement signals.

The VM approach does not require such a basis function. Unfortunately, it is not able to detect the faster signal without trying each possible time delay for both signals. This results in a computationally expensive brute force calculation in the first time step. Afterwards, it is very efficient because it must only execute basic math operations and searches only for a restricted number of possible delays. Compared to the other approaches, VM requires the smallest frame size to retrieve feasible results. In total, Table 6 shows clearly that VM is the most accurate and fastest of the three proposed approaches with the lowest memory consumption.

For the high precision and low run-time, we decide to continue our work with the VM concept. Before we are able to apply our approach to series production EVs, we require further optimization as shown in Section 3.4 to stabilize the estimated time delay. This stabilization comes with another drawback. The algorithm requires more time steps to pass before it adapts to a new delay. Nevertheless, regarding that succeeding power train control functions require stable inputs, this drawback seems acceptable for us. Another drawback of the optimization is the threshold value *K* in Equation (Equation 15) which must be chosen manually. Although it does not require expert knowledge but can be set by trial and error, we would prefer an automated way for finding the optimal value for *K*.

## 7. Conclusions and Outlook

This article presents three different approaches for TDE of measurement signals in the power train of EVs. As automotive ECUs are designed very efficiently, our evaluation’s focus lies also on computation and memory complexity and not solely on accuracy. Unfortunately, LR is not suited for our purposes because it suffers too much from vertical offsets in the measurement data. However, with VM, we present a feasible approach for TDE of distributed sensor systems of EVs. AFs are also not suited because they require too large frame sizes and have lower accuracy than VM. We recommend using VM due to its high estimation accuracy and computational efficiency. As the output of VM is not stable enough to directly process it to power train control functions of series EVs, we optimize it first. For the optimization, we introduce a threshold value as additional requirement for changing the value of the estimated time delay. The new requirement decelerates the detection of changed time delays. Nevertheless it improves the TDE’s stability and accuracy.

After the introduction of an automated TDE system, we now know each signal’s delay. However, if we correct the delay, we move some signals to the past and lose the measurements corresponding to the latest time steps. This is correct because in fact we do not receive up-to-date measurements, only delayed ones from the past. We really do miss the last measurements and there is a gap between the last received measurement and the present time step. Thus, our next work focuses on possible ways to close this gap by replacing the hidden information about the missing measurements from the latest time steps.

## 8. Patents

The TDE for the power trains of EVs is registered at the German Patent and Trade Mark Office (DPMA) as patent application. Both the VM approach as well as its RMSE-based version for TDE in the power trains of EVs are registered there as a common patent application. The optimization is registered as a third patent application resulting from this work.

## Figures and Tables

**Figure 1 sensors-20-00351-f001:**
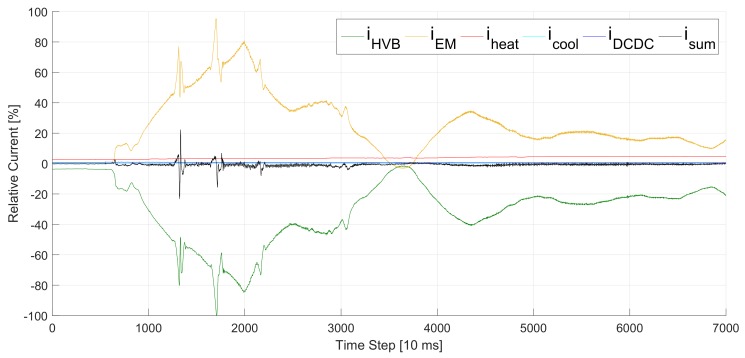
Currents of all HV components in an EV on a test drive. The sum of all currents *i*_sum_ is plotted in black. According to *Kirchhoff’s current law*, it should be constantly 0 %. However, looking at the measurements shows that the deviation *i*_sum_ is higher than the current of the DCDC converter *i*_DCDC_. Even its noise spectrum is approximately half as high as the consumption of the heating *i*_heat_, which is the second largest consumer in this drive.

**Figure 2 sensors-20-00351-f002:**
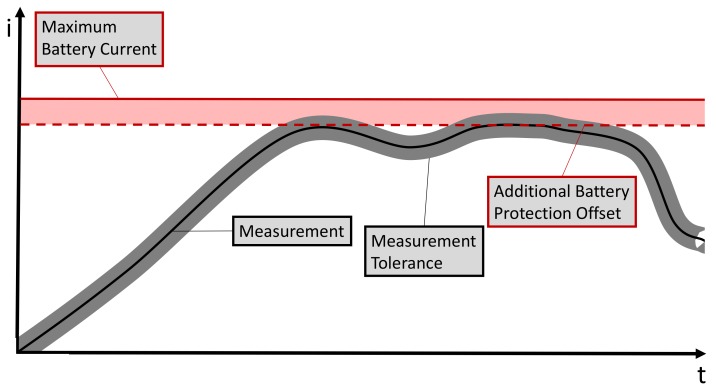
A simplified example of offsets for protection of the HVB. The measured value (black) differs from the real value in the range of some tolerance (grey). To prevent exceeding the battery limit (red, solid) even under the worst measurement conditions, an additional battery protection offset (red, dashed) is introduced. The same principle is used analogously with negative currents. It can be extended to other HV components.

**Figure 3 sensors-20-00351-f003:**
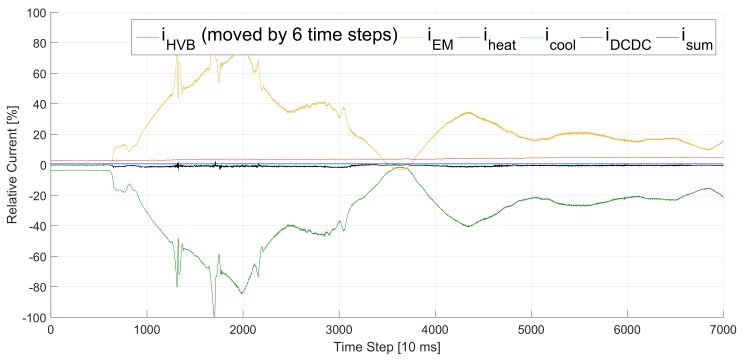
The same test drive as in Figure 1 but with the battery current *i*_HVB_ (green) shifted by six time steps. The sum of all currents *i*_sum_ (black) is significantly closer to 0 %.

**Figure 4 sensors-20-00351-f004:**
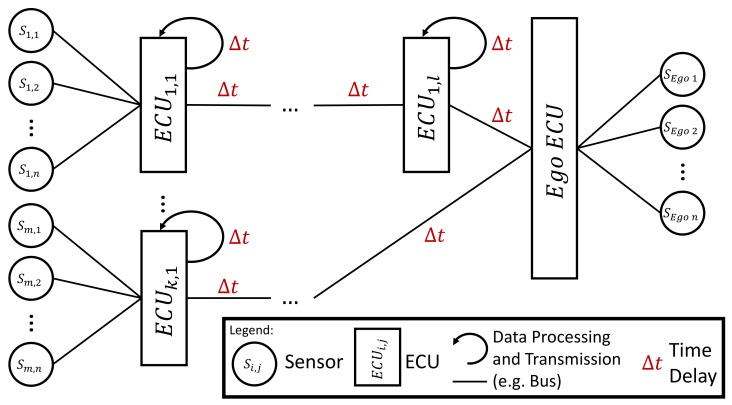
A schematic example of an automotive bus system with higlighted sources of time delays. Please note that the time delays are highly individual and not necessarily equal, but constant or only slowly changing. The ECUs can be connected directly or indirectly via other ECUs. The Ego ECU is not able to reconstruct the time delays, because it only knows the received measurement values and their last sender. It has no further information about the time passed since the measurement’s original creation.

**Figure 5 sensors-20-00351-f005:**
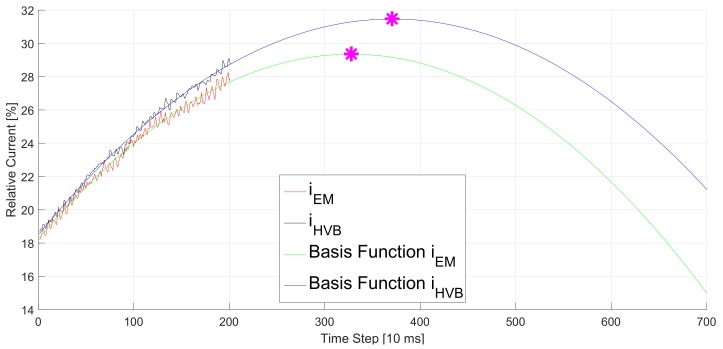
Basis functions of *i*_HVB_ (black) and *i*_EM_ (red) simulated by LR (blue and green, respectively). The magenta marked points are the vertexes. Their horizontal difference is 43 time steps in contrast to the real time delay which is 6 time steps. The wrong TDE is caused by the noise and the vertical offset of the measurement signals.

**Figure 6 sensors-20-00351-f006:**
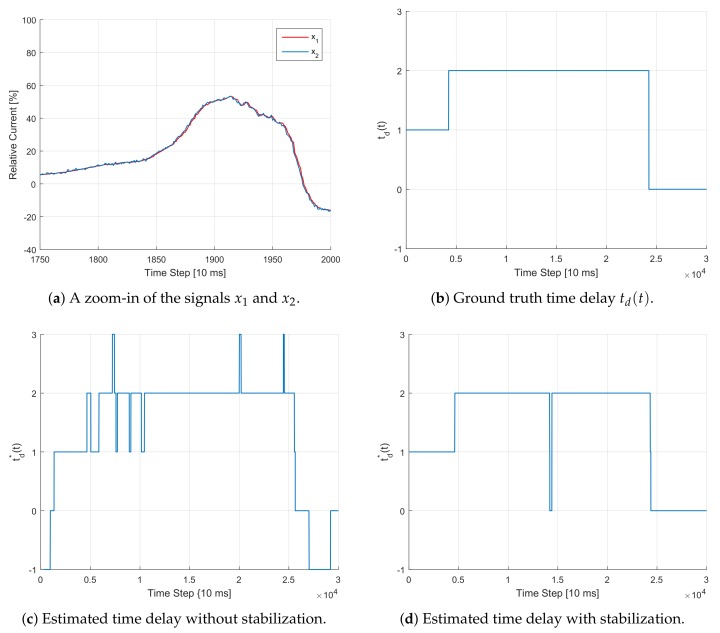
Illustration of the TDE estimation procedure using the VM approach with and without stabilization for the case of simulated signals. Clearly, the estimation is more stable when using the criterion in Equation (Equation 15). Waiting for the *right* moment to perform an update comes however with the expense of a slightly delayed, yet more reliable, prediction. For example, the jump of *td(t)* from 1 to 2 was detected with a delay of around 320 steps, which corresponds to around 3.2 seconds.

**Figure 7 sensors-20-00351-f007:**
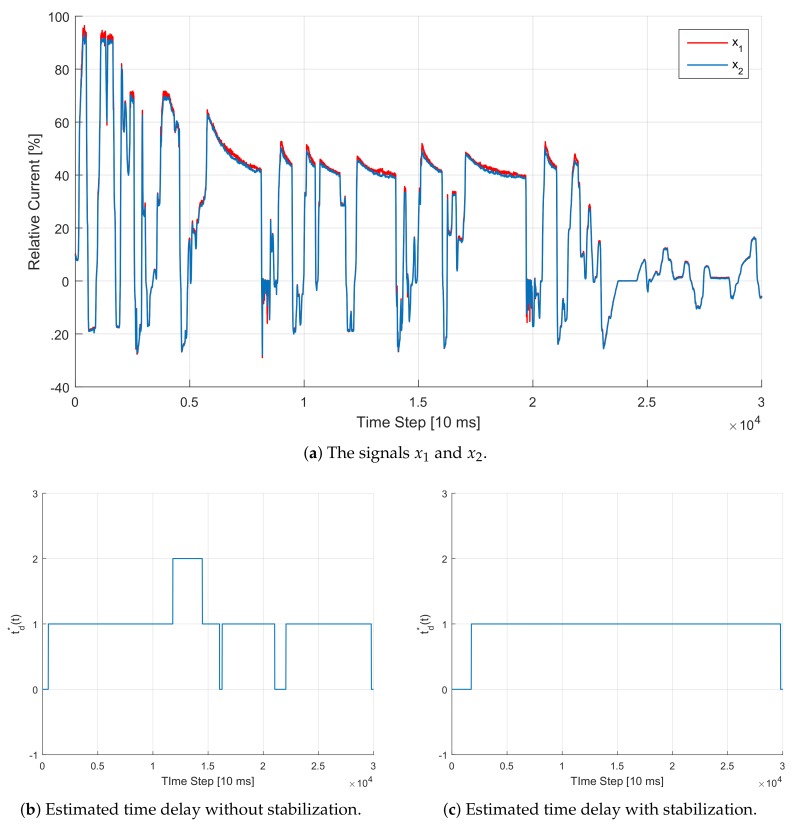
Illustration of the TDE estimation procedure using the VM approach with and without stabilization for the case of real signals. Although we do not know the underlying true time delay, it is again clear that the TDE is more stable using the suggested approach.

**Table 1 sensors-20-00351-t001:** RMSE and run-time analysis of the AF concept for different algorithms with a frame size of $2^{10}$.

	RMSE	Average Run-Time (s)
**LMS**	2.8972	$1.13\;\cdot \;10^{-2}$
**NLMS**	2.5163	$1.31\;\cdot \;10^{-2}$
**LMF**	2.6138	$1.14\;\cdot \;10^{-2}$
**RLS**	2.276	$1.07\;\cdot \;10^{-2}$

**Table 2 sensors-20-00351-t002:** RMSE and run-time analysis of LMS with different learning rates and a frame size of $2^{10}$.

	RMSE	Average Run-Time (s)
a=0.01	3.0451	$1.14\;\cdot \;10^{-2}$
a=0.05	2.3608	$1.14\;\cdot \;10^{-2}$
a=0.1	2.1479	$1.14\;\cdot \;10^{-2}$
a=0.5	2.6484	$1.14\;\cdot \;10^{-2}$
a=1	3.2474	$1.14\;\cdot \;10^{-2}$

**Table 3 sensors-20-00351-t003:** RMSE and run-time analysis of the LMS algorithm with a shorter frame size of $2^{8}$.

	RMSE	Average Run-Time (s)
a=0.01	2.8603	$2.80\;\cdot \;10^{-3}$
a=0.05	2.5598	$2.80\;\cdot \;10^{-3}$
a=0.1	2.3994	$2.80\;\cdot \;10^{-3}$
a=0.5	2.4983	$2.80\;\cdot \;10^{-3}$
a=1	3.0781	$2.80\;\cdot \;10^{-3}$

**Table 4 sensors-20-00351-t004:** RMSE and run-time analysis of the LR concept.

	RMSE	Average Run-Time (s)
**Frame Size 30**	$5.83\;\cdot \;10^{10}$	$1.74\;\cdot \;10^{-4}$
**Frame Size 200**	$4.92\;\cdot \;10^{4}$	$9.80\;\cdot \;10^{-4}$
**Frame Size 300**	$4.47\;\cdot \;10^{4}$	$1.40\;\cdot \;10^{-3}$

**Table 5 sensors-20-00351-t005:** RMSE and run-time analysis of the VM concept.

	RMSE	Average Run-Time (s)
**Frame Size 30**	2.0696	$4.54\;\cdot \;10^{-5}$
**Frame Size 50**	1.3034	$4.70\;\cdot \;10^{-5}$
**Frame Size 100**	1.2364	$4.77\;\cdot \;10^{-5}$
**Frame Size 200**	1.2215	$5.16\;\cdot \;10^{-5}$
**Frame Size 300**	1.1825	$5.79\;\cdot \;10^{-5}$

**Table 6 sensors-20-00351-t006:** RMSE, run-time analysis and frame size of all three concepts compared to each other.

	RMSE	Average Run-Time (s)	Frame Size
**Adaptive Filter**	2.3994	$2.80\times 10^{-3}$	$2^{8}$
**Linear Regression**	$4.92\times 10^{4}$	$9.80\times 10^{-4}$	200
**Variance Minimization**	1.3034	$4.70\times 10^{-5}$	50

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
