# Peer review of "Evaluation of Three Different Approaches for Automated Time Delay Estimation for Distributed Sensor Systems of Electric Vehicles"

_sensors, 2020, doi:10.3390/s20020351_

Round 1

Reviewer 1 Report

The article presents a comparison between three approaches for automated time delay estimation (TDE) in electric vehicle powertrains, namely linear regression, variance minimization and adaptive filters. The contribution is timely and highly relevant to the scope of the Sensors journal given its topic.

Some minor aspects would need to be revised as follows:

in the abstract when discussing the three methods, the interested reader would benefit if some quantitative indicator of the performance of the best method would be given e.g. "we find that the best method offers a xx% improvement over..." in the introduction before technical details regarding measurement errors in electrical circuits, it would be required a broader descriptive paragraph on the context and justification of the work e.g. brief description of the key measurement aspects of EV powertrains and highlighting the area that the current article addresses; the state-of-the-art as reflected by the references list should be expanded by adding 5-10 more recent and relevant journal publications, with particular focus on related work that fits the scope of the Sensors journal; subsection 3.3. is poor and too short, at least the formulas/relationships for the chosen adaptive filter implementation would be given - not only referring the reader to reference [7]; the results would be more easily followed if acronyms for the three methods would be introduced e.g. LR - linear regression, VM - variance minimization and AF - adaptive filtering; a joint table is required at the end to compare directly the three methods. 

Author Response

Dear Reviewer,

thank you so much for your constructive feedback. We considered your points as follows:

Abstract: We added the sentence: "By correcting these delays with accurate Time Delay Estimation (TDE), our data shows that we can reduce the measurement deviations from 25 % of the maximum current to below 5 %." Intoduction: We added a new paragraph right in the beginning and a new figure. List of References/State of the Art: We increased the cited references from 9 to 22.
19 references now cited in the State of the Art.
12 journal papers + 2 proccedings + 2 books cited in total.
4 papers from "Sensors" + 2 from the related journal "Energies" cited in total. Subsection 3.3: We wrote the whole subsection new. Acronyms: We introduced the three proposed acronyms. Joint table with results of all 3 concepts: We added Table 6.

Best regards and a merry Christmas!

Jakob Pfeiffer

Reviewer 2 Report

Dear Authors

Thank you for taking into account my previous comments.

I have some comments on your article:

The literature review is still poor. It contains only 9 items. The number of references cited should be yet increased. Section 2. State of the Art - should be revised after adding new literature items.

Author Response

Dear reviewer,

thank you very much for your feedback.

We increased the cited references from 9 to 22.
19 references now cited in the State of the Art.
12 journal papers + 2 proccedings + 2 books cited in total.
4 papers from "Sensors" + 2 from the related journal "Energies" cited in total.

Best regards and a merry Christmas,

Jakob Pfeiffer

This manuscript is a resubmission of an earlier submission. The following is a list of the peer review reports and author responses from that submission.

Round 1

Reviewer 1 Report

The paper describes the authors' investigation in terms of estimating the time delay of different electrical components in order to measure the current signal more precisely to better avoid the damage of HVB due to imprecise current measurement results. The authors are suggested to add some explanation in part 3 to improve the paper's readability.

Reviewer 2 Report

The authors are presenting the automated system for measuring of signal delays which is used for the calculation of the real currents in EV.

In my opinion, the main two questions which should be explained in the paper are the following:

If the system properly identifies the delay of different current signals, the individual and the overall actual currents can be estimated properly. However, those identified “actual” currents are not the actual (“current”) electric currents, but the delayed currents, which are delayed by at least the max. identified delay td,i (plus time for computation, etc.). How does this delay influence performance of the power limitation algorithms in EV? In variance minimization algorithm, it is written that E (7) should be applied in expression (8). Let us consider the case where we have, in some time period, linearly increasing signal x1: x1=t and 1 sample delayed signal x2: x2=t-1. From (7), E=1. If we calculate variance (8), the variance is already zero, so the optimal td,i = 0, which is not correct. The authors should explain in which cases (for what kind of measured signals) the algorithm works and when it fails. Without this information, it is hard to evaluate the effectiveness of the proposed method.

Reviewer 3 Report

Dear Authors,

I have some comments on your article:

General remarks:

The literature review is too poor. Contains only 5 items. The number of references cited should be significantly increased. It seems that the article does not have enough information on how to collect data, especially in the aspect of their simultaneous measure and collection. Please provide more information on how your data is processed by the on-board signal analysis system and time reduction method delay in obtaining the required signal samples.

Detailed comments:

Please check all equation, symbols, and index used in the text and equations. Section 2. State of the Art - should be extended after adding new literature items. It seems to me that all obtained results and references in the content of the article should be described in an impersonal form. I would rather not use phrases, for example, we realize - rather: realize. Section 4.1. Data – the article should contain more information on how to collect data and the used sensors. Descriptions of the tables. The table's description should be above and not below the tables. Section Abbreviations - the acronyms should rather be written in capital letters, e.g. electronic control unit, I think it should be: Electronic Control Unit.